# On the complex structural diffusion of proton holes in nanoconfined alkaline solutions within slit pores

Daniel Muñoz-Santiburcio[1] & Dominik Marx[1]

The hydroxide anion $OH^-$(aq) in homogeneous bulk water, that is, the solvated proton hole, is known to feature peculiar properties compared with excess protons solvated therein. In this work, it is disclosed that nanoconfinement of such alkaline aqueous solutions strongly affects the key structural and dynamical properties of $OH^-$(aq) compared with the bulk limit. The combined effect of the preferred hypercoordinated solvation pattern of $OH^-$(aq), its preferred perpendicular orientation relative to the confining surfaces, the pronounced layering of nanoconfined water and the topology of the hydrogen bond network required for proton hole transfer lead to major changes of the charge transport mechanism, in such a way that the proton hole migration mechanism depends exquisitely on the width of the confined space that hosts the water film. Moreover, the anionic Zundel complex, which is of transient nature in homogeneous bulk solutions, can be dynamically trapped as a shallow intermediate species by suitable nanoconfinement conditions.

[1] Lehrstuhl für Theoretische Chemie, Ruhr-Universität Bochum, 44780 Bochum, Germany. Correspondence and requests for materials should be addressed to D.M.-S. (email: daniel.munoz@theochem.rub.de).

The solvated hydroxide ion in water offers surprising features, both structural and dynamical, compared with what has been expected for a long time according to the naive 'mirror image' picture of the hydrated excess proton[1]. Based on extensive simulation work and subsequent experimental confirmation, it may be said that $OH^-$ (aq) is well understood in the limits of both bulk solvation and microsolvation[1]. However, the solvated hydroxide in inhomogeneous aqueous environments is still far from being understood, resulting in an ongoing and lively debate revolving around the properties of water's two autoprotolysis products, $H^+$ (aq) and $OH^-$ (aq), in the context of interfacial water. One of the most elementary questions in this respect, 'Is the water surface acidic or basic?', is controversially discussed by different authors[2–6]. Although most studies report that water surfaces at water/air and water/oil interfaces are negatively charged[7–10], quite different explanations for this have been proposed, several of them actually involving hydroxide ions, and there is also no consensus yet regarding the origin of this surface charge[11–19]. A summary of the different, mostly conflicting results—both from experiments and simulations—is provided in the introduction of ref. 20 including substantial referencing of earlier work. Clearly, more studies are needed to eventually settle this longstanding debate.

In the broader context of interfacial solutions beyond the simple water/vapour system, much interest in the $OH^-$ (aq) species in confined environments is arising, for instance as a consequence of recent developments in anionic exchange membranes[21,22], which are gaining attention for their use in electrochemical devices such as fuel cells as an alternative to the more traditional proton exchange membranes. Especially alkaline aqueous solutions in reduced dimensionality and nanoconfinement are becoming highly relevant in view of the very distinct properties of nanoconfined water compared with the bulk. Thus, $OH^-$ (aq) has already been carefully studied in nanoconfined water wires[23,24] and monolayer films[25]. In the realm of confinement, it is stressed that computer simulation is considered to be a valuable complement to experiment in view of challenges encountered when probing experimentally the acidic/basic character of interfacial versus bulk-like water in nanoconfinement[26].

In this work, we uncover particularly surprising dynamical properties of $OH^-$ (aq) in nanoconfined water layers between mackinawite FeS sheets based on *ab initio* molecular dynamics simulations[27] in conjunction with an elaborate model of this mineral-based slit pore. Mackinawite has been proposed as a putative nanoreactor and possibly even catalyst for prebiotic chemistry at elevated thermodynamic conditions close to deep-sea hydrothermal vents[28], where high temperatures and pressures of typically $\approx 500\,K$ and $\approx 20\,MPa$, respectively, are encountered. Its layered structure can be intercalated by water and it has been suggested that a primordial 'pyrophosphate synthetase nanoengine' could have emerged, thanks to the charge gradients that can be established along these nanochannels[29,30]. Because of these ramifications and as a prologue for an upcoming study of prebiotic reactions[31,32] in such environments, we set out to investigate the properties of nanoconfined water in mackinawite at the relevant elevated temperature and pressure conditions[33], and also those of the solvated excess proton therein[34]. Transcending previous work, we compare two different confinements of alkaline aqueous solutions in an overall realistic setup, one with extreme confinement where water forms a monolayer and another one where the aqueous phase is a water bilayer. Anticipating our core result, it is shown that these confinements imprint stark and unexpected differences on the structural diffusion mechanism of $OH^-$ (aq) unknown from $H^+$ (aq).

## Results

**General features of nanoconfined alkaline water lamellae.** We have studied $OH^-$ (aq) in nanoconfined water between mackinawite sheets using our well-validated setups[33,34], namely the systems *N* (narrow slit pore, Fig. 1) and *W* (wide slit pore). In extreme confinement, system *N*, where water forms a single-layer hydrogen-bonded network (Fig. 2 top), the dynamics of the basic solution is not liquid-like, as the water molecules are 'arrested' at certain preferred positions, which is consistent with what was observed for both neutral[33] and acidic[34] water lamellae. On the other hand, the most interesting aspect in system *W* is the stratified structure of the alkaline aqueous phase, where a water bilayer is clearly formed (Fig. 2 bottom and Fig. 3). The water dynamics in this case is liquid-like and again this is consistent with our previous simulations of neutral and acidic nanoconfined water[33,34].

**Structure and orientation of the $OH^-$ (aq) complex.** In the homogeneous bulk water environment, $OH^-$ is known to preferentially accept four hydrogen bonds in a square-planar arrangement, while H′ donates none, that is, $OH^-(H_2O)_4$ as depicted in Fig. 4f including standard notation, which is called the 'hypercoordinated' resting or majority state according to the dynamical hypercoordination mechanism[1] of hydroxide structural diffusion in the bulk limit. When one of these accepted hydrogen bonds is broken because of suitable thermal fluctuations of the hydrogen bond network and $OH^-$ is transiently donating a hydrogen bond via its hydrogen H′, then the $OH^-$ is properly 'presolvated'[35] in a topology similar to the ideal tetrahedral coordination of a $H_2O$ molecule in bulk water. In such favourable configurations, proton hole transfer along the most active accepted hydrogen bond, that is, $O^\star \cdots H^\star\text{–}\tilde{O}$, readily occurs to $O^\star$, thus forming a water molecule $H'H^\star O^\star$ in its ideal solvation structure. This leads to the concurrent displacement of the charge defect to the previous first

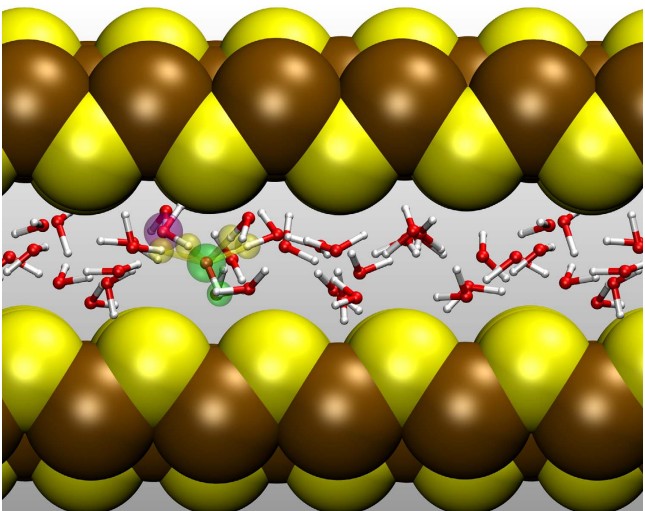

**Figure 1 | Solvated $OH^-$ in the narrow slit pore system *N*.** The mackinawite Fe and S atoms are shown as large brown and yellow spheres, whereas O and H atoms are represented in terms of red and white balls-and-sticks, respectively. The $OH^-$ defect is in its hypercoordinated resting state and assumes the preferred perpendicular orientation of its O–H axis with respect to the mineral surface as highlighted using large green spheres for $O^\star$ and H′; the four H bonds accepted by $O^\star$ are distinguished using yellow spheres and the $\tilde{O}$ atom is marked by a violet sphere; see Fig. 4f for atom labelling.

neighbour site $\tilde{O}$, which quantitatively explains the full proton hole transfer kinetics[36,37].

In the inhomogeneous, nanoconfined water environment, the first striking feature of $OH^-$(aq) can be extracted from the number density profiles normal to the water/mineral interface (Fig. 2) together with the joint probability of the position and angle of the $OH^-$ unit relative to the confining solid surfaces (Fig. 5a), which discloses pronounced spatial and orientational preferences. The results for the narrow pore, system $N$, agree with similar calculations for solvated $OH^-$ between graphene slabs[25]; we also find the O–H$^-$ unit to be oriented perpendicular to the confining surfaces and coordinated to four water molecules (Fig. 1). In our case, the 'eggbox' corrugation of the mackinawite surface imprints a superstructure on the nanoconfined water monolayer and the $OH^-$ unit is displaced towards the midplane, which minimizes the distortion of the square-planar $OH^-(H_2O)_4$ complexes.

In stark contrast, the profiles for the wide system $W$ show remarkable peculiarities. Here, the $OH^-$ is nicely integrated into either one of the water layers, but the only hydrogen of the $OH^-$, H′, is preferentially located close to the confining surface instead of being buried in the water bilayer. Close inspection of the trajectories reveals that the $OH^-$ anion is in about half of the time fourfold coordinated in a square-planar configuration, where all the waters of the first hydration shell are in the same water layer, but oriented such that the H′ atom points to the confining mineral surface as depicted in Fig. 3a. Other possible configurations—yet less probable – feature the $OH^-$ pointing to the neighbouring water layer where it gets 'buried' (Fig. 3b), or being 'tilted' such that its first solvation shell belongs to both water layers (Fig. 3c). The relative probability of these configurations can be easily extracted quantitatively from Fig. 5a.

**Proton hole transfer and structural diffusion mechanisms**. The free-energy profiles for proton hole transfer (Fig. 6a), as obtained by considering the well-established generalized coordinate $\delta = d(O^* - H^*) - d(\tilde{O} - H^*)$, yield barriers of roughly $0.5\ \mathrm{kcal\ mol^{-1}}$ ($\approx (1/2)k_B T_{500}$) in both systems, $N$ and $W$, which is slightly above the corresponding barrier in bulk water. However, is the proton hole migration mechanism in nanoconfined water lamellae the same as in the homogeneous bulk environment? In the narrow system $N$, which is characterized by a single water layer, $OH^-$(aq) is preferentially coordinated to four water molecules and, moreover, most probably points with its H′ towards either the 'upper' or 'lower' confining surface—being key to charge migration. This so-called 'exposed' ($E$) interfacial state in the inhomogeneous environment corresponds to the hypercoordinated resting state with respect to proton migration as depicted by the leftmost structure of Fig. 4a and indicated by high probabilities in Fig. 5. Next, the loss of one of its four hydrogen bonds leaves the $OH^-$ accepting three hydrogen bonds. This state, on proton transfer of H$^*$ from a water molecule H$^*$H$\tilde{O}$ in the first solvation shell (see Fig. 4f for site labelling), results into the formation of a water molecule H′H$^*$O$^*$ that is accepting two hydrogen bonds and donating one (to $\tilde{O}$), which agrees substantially with recent simulations of an alkaline water monolayer between graphene sheets[25].

The striking difference with hydroxide migration in bulk water[1] is that it is not possible for the nascent water molecule, H′H$^*$O$^*$, being necessarily located right at the interface, to achieve the ideal tetrahedral coordination with two donor and two acceptor hydrogen bonds. However, importantly, the O$^*$H′ bond of the nascent water molecule is a 'dangling' OH bond (also called free OH or single-donor species), which is one out of several ideal hydrogen bond arrangements that can terminate planar water interfaces[38]. Indeed, there is now solid evidence accumulated that the preferred termination of the water surface in contact with vapour occurs via such dangling OH bonds, which implies that these water molecules donate only a single hydrogen bond towards the interior[39]. This is exactly the situation met in the centre panel of Fig. 4a, where this donated

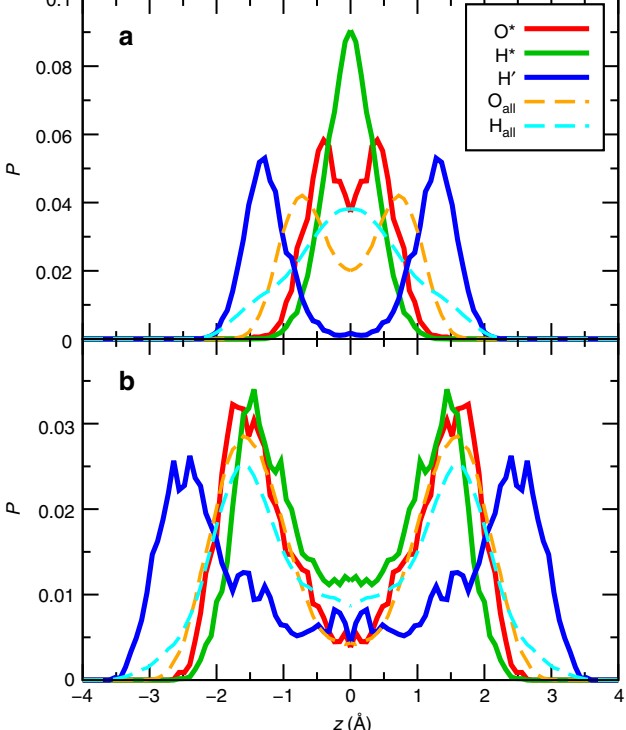

**Figure 2 | Normalized and symmetrized number density profiles.** Normalized and symmetrized number density profiles perpendicular to the mineral surface for the narrow ($N$, **a**) and wide ($W$, **b**) systems averaged over all O and H sites, and resolved specifically for the O$^*$, H$^*$ and H′ sites as defined in Fig. 4f.

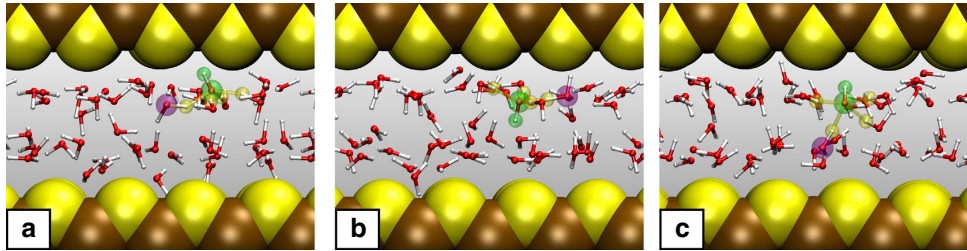

**Figure 3 | Snapshots depicting different representative configurations of $OH^-$ (aq) in the wide slit pore.** System $W$, where **a** covers the 'exposed' state E, as well as the 'buried' and 'tilted' states B/T in **b** and **c**, respectively; see Fig. 5 for definitions.

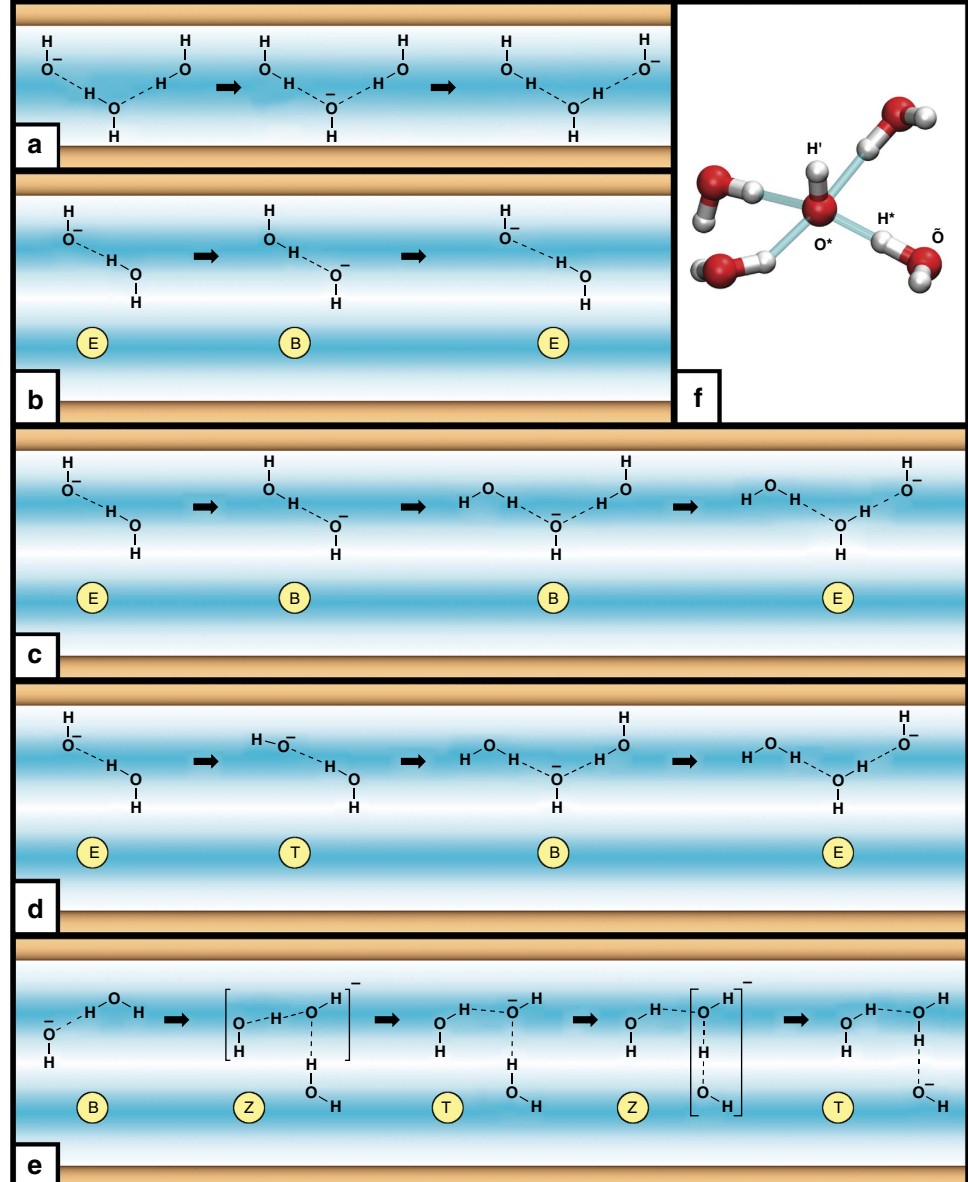

**Figure 4 | Schematic representation of the distinct proton hole migration mechanisms.** Schematic representation of the distinct proton hole migration mechanisms in the narrow and wide slit pores, that is, system $N$ in (**a**) and system $W$ in (**b**-**e**). The top and bottom confining surfaces are schematically included as brown bars, whereas the high/low-density water regions are represented using light/dark blue shading, to visualize the water monolayer/bilayer structure of the water film in system $N/W$, respectively. Only key species and hydrogen bonds (dotted lines) are shown in terms of chemical formulae, thus omitting the full solvation shell of the $OH^-$ defect (with the negative charge symbol attached to it, see also **f**). In the narrow pore limit, the preferred zig–zag migration path of $OH^-$ is depicted in **a**. In the bilayer pore $W$, one can distinguish proton (hole) rattling relative to the 'exposed' state $E$ without net charge migration (**b**), structural diffusion via the 'exposed'–'buried'–'exposed' $E$–$B$–$E$ (**c**) and 'tilted'–'buried'–'exposed' $T$–$B$–$E$ (**d**) mechanisms, as well as structural diffusion via alternating 'tilted' and 'buried' configurations (**e**), which stabilizes the trapped Zundel intermediate $Z$ (see text) as symbolized in **e** using square brackets; see Fig. 5 for definitions. The standard notation employed for the solvated $OH^-$ (aq) complex is depicted in **f** using a representative snapshot of the preferred hypercoordinated resting state as sampled at the interface. Here, $O^*$ and $H'$ denote the site of the hydroxide anion, whereas $H^*$ and $\tilde{O}$ are the sites of the particular water molecule, which donates the most active hydrogen bond along which proton (hole) transfer will eventually occur.

hydrogen bond is accepted by the nascent $OH^-$ defect after proton hole transfer. Thus, taking that viewpoint, it is evident that not being able to form the weak donor hydrogen bond via $H'$, which ensures fourfold hydrogen bonding of the nascent water $H'H^*O^*$ as observed in bulk migration processes[1], should not block the proton transfer event towards $O^*$ if that occurs right at an interface, in full accord with the presolvation concept[35]. Moreover, our reference calculations of $OH^-$ in bulk water (see 'Methods' section) showed a further weakening of the donor

bond via $H'$ at the relevant high temperature and pressure conditions compared with ambient, while still adhering to the dynamical hypercoordination mechanism[1] of hydroxide structural diffusion. This supports our observation that the absence of this bond should not hinder proton hole transfer in nanoconfined water at these elevated conditions.

Yet, there is an intricate geometric constraint that is imposed by the planar monolayer confinement in the narrow system $N$. This comes because the water molecule that turns into the new

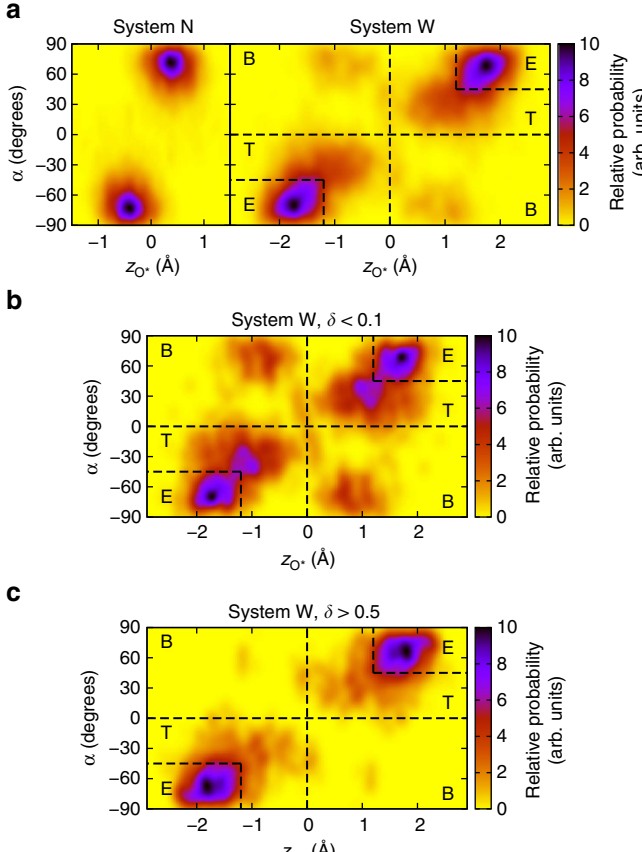

**Figure 5 | Analysis of the orientations of the molecular axis of OH⁻ (aq).**
(**a**) Joint probability distributions (symmetrized w.r.t. $z = 0$) as a function of
the angle $\alpha$ between the O*–H′ bond axis and the $xy$ plane, and the distance
$z_{O*}$ of O* w.r.t. the slit pore midplane for the narrow ($N$, left panel) and wide
($W$, right panel) systems. (**b,c**) For system $W$, the total distribution function
from **a**/right is respectively dissected in terms of configurations in the
active state close to transfer events ($\delta < 0.1$) and in those corresponding to
the hypercoordinated resting state ($\delta > 0.5$) where the centred hydrogen
bond characterizes the anionic Zundel complex $Z$, $[\mathrm{HO} \cdots \mathrm{H} \cdots \mathrm{OH}]^-$;
see text for definition of the transfer coordinate $\delta$. Concerning orientation
and location of the charge defect in the upper layer $z_O* > 0$ of the wide
pore $W$, the so-called 'exposed' states $E$ are those where $\alpha > 45°$
together with $z_O* > 1.2$ Å and thus represent situations where the
$H$ site of interfacial OH⁻ species points towards the upper surface,
while the O–H axis is roughly perpendicular to that surface. The 'tilted'
states $T$ close to the upper surface are those where $\alpha > 0$ but excluding the
$E$ states, whereas the 'buried' states $B$ are those where $\alpha > 0$. The
corresponding definitions for the 'lower' surface, $z_O* < 0$ Å, follow from
symmetry (see also Fig. 4).

OH⁻(aq), that is, $\tilde{\mathrm{O}}\mathrm{H}^-$, must be oriented such that its O–H axis
is also perpendicular to the surface. Otherwise, the formation of
the preferred resting state would not be possible and the very
same proton that has just been transferred would immediately
rattle back, thus reforming the previous OH⁻, that is, O*H′ and
therefore not leading to any charge transport. Indeed, these
rattling events in system $N$ are very frequent and it has been
suggested that the additional time necessary for reaching this
proper arrangement of the proton donating water molecule is the
reason for the slower diffusion rate of OH⁻(aq) in monolayers[25]
compared with the bulk. In addition to that, we observe in our
simulations that the nascent OH⁻(aq) will point overwhelmingly
with its H′ towards the opposite surface of that pore, see Fig. 4a.

This phenomenon is the result of an intricate interplay of extreme
nanoconfinement in the monolayer limit, the preferred
orientation of interfacial OH⁻ species and the essentially
tetrahedral topology of the hydrogen bond network. The net
outcome is that proton hole transfer in the narrow slit pore limit
preferentially follows a zig-zag migration path where OH⁻ jumps
between the two opposing planar surfaces, thereby reorienting its
dipole moment in each such step as illustrated by Fig. 4a. This
scenario is fully consistent with the respective statistical
analyses in Figs 5a and 6a.

In the wide slit pore $W$, the mechanism for proton hole transfer
and subsequent structural diffusion is found to be even more
intricate as a consequence of the bilayer sub-structure of the water
lamella. The hypercoordinated resting state of the OH⁻(aq)
complex at the interface is of course identical to that in the $N$
system (Fig. 3a), which is the 'exposed' configuration $E$. As in the
previous case, the loss of one of these hydrogen bonds leaves the
OH⁻ in a properly presolvated state that, on proton transfer
from one of the surrounding waters, forms an interfacial
'single-donor' water molecule that donates one hydrogen bond
and accepts two. Now, the key distinction is that the nascent
OH⁻ almost always finds itself in a qualitatively different
solvation environment from the narrow pore scenario: either this
new OH⁻ species is pointing with its H′ hydrogen towards the
other water layer of the bilayer film as depicted in Fig. 3b, which
is what we call the 'buried' configuration $B$, or it is in a 'tilted'
situation $T$ as illustrated by Fig. 3c; see Fig. 5 for the definitions
of the E/B/T states. Both the 'buried' and 'tilted' arrangements make
it easy for the nascent OH⁻ unit to reach the ideal tetrahedral
solvation state of a water molecule in bulk. Hence, the newly
formed OH⁻ is prone to receive a proton via any of the hydrogen
bonds that are donated to it, which is however most likely to be
the same hydrogen bond along which the previous proton
transfer took place! This leads to confinement-induced rattling as
visualized schematically in Fig. 4b and corresponds to inter-
conversions between Fig. 3a–c without any net charge transport.

The striking difference between the initial 'exposed' state and
the most unstable 'tilted'/'buried' configurations is clearly
revealed if we analyse separately these limiting cases in terms of
the joint probabilities as depicted in Fig. 5b,c. This analysis makes
clear that the resting state with respect to proton transfer
(that is, $\delta > 0.5$) coincides with OH⁻(aq) being 'exposed',
whereas active states with centred hydrogen bonds ($\delta > 0.1$) go
hand in hand with OH⁻, finding itself in either 'buried' or 'tilted'
configurations. This structural cross-correlation can be
immediately linked to free-energy profiles for proton hole
transfer once separated in terms of the 'exposed' and
'tilted'/'buried' scenarios (Fig. 6a). The proton hole transfer
barrier involving only 'exposed' states is found to be dramatically
higher, by roughly a factor of 4, relative to those situations where
OH⁻(aq) is initially in either a 'buried' or 'tilted' arrangement.
This facile proton transfer for $T/B$ configurations in the wide slit
pore correlates nicely with formation of the donated hydrogen
bond via H′ in active complexes ($\delta < 0.1$), which on detailed
analysis of conditional radial distribution functions together with
their running coordination numbers is revealed to be more
pronounced than in the corresponding bulk regime (whereas this
H′ bond is essentially absent both in $E$ configurations in the $W$
system, as well as in the narrow pore confinement). We infer that,
although the presence of this bond donated by OH⁻ is not
strictly necessary for proton (hole) transfer to occur at all, it
greatly facilitates this process when the structural constraints
imprinted by the confinement allow for or even favour its
formation.

These different solvation environments, which can be reached
on proton hole transfer from 'exposed' to 'buried'/'tilted'

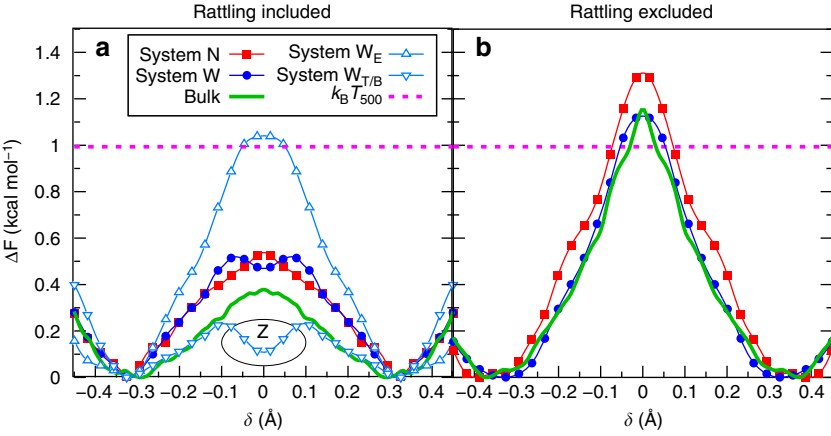

**Figure 6 | Free-energy profiles for proton hole transfer in the narrow and wide slit pores.** Free-energy profiles for proton hole transfer in the narrow and wide slit pores compared with the homogeneous bulk environment at the same thermodynamic conditions as a function of the transfer coordinate $\delta$ including and excluding proton hole rattling events (see text) in **a** and **b**, respectively. The free-energy signature of the trapped Zundel intermediate $Z$ is highlighted using a black ellipse and the total free-energy profile in the wide pore is additionally split into contributions due to $E$ and $T/B$ states. The definitions of the $E$, $T$, $B$ and $Z$ charge defect states are provided in Fig. 5 (see also Fig. 4) and the horizontal dashed line marks the corresponding thermal energy.

configurations and *vice versa* only in the wide pore, are the reason of frequent proton rattling events. In this case, a specific $OH^-$ receives and back transfers a proton involving exclusively water molecules in its first solvation shell, see Fig. 4b. Hence, for structural diffusion and thus charge migration to happen, some additional condition must be met that avoids the reformation of the initial $OH^-$ unit. On carefully analysing our trajectories, we have identified three distinct events that can avoid this: (i) a proton hole transfer event is followed by the reorientation of the nascent water molecule so the eventual rattling of the just transferred proton would not lead to reformation of an $OH^-$ species pointing to the surface (as illustrated by the mechanism in Fig. 4c); (ii) the $OH^-$ unit itself reorients so that the H′ points away from the confining surface, which can happen via changes in the first solvation shell or vehicular (Stokes) diffusion of the entire complex caused by thermal fluctuation (Fig. 4d); (iii) a change in the second solvation shell makes possible fast Grotthuss-like proton transfer across a water wire so the newly formed $OH^-$ is no longer the first neighbour of the former resting state, thus suppressing the probability of a rattling event that will lead to its reformation (not depicted).

It is observed that structural diffusion in the wide slit pore mostly occurs via mechanisms (i) or (ii): once the preferred orientation of the former O★H′ bond is lost, the formation of the same initial 'exposed' state after proton rattling is no longer possible and subsequent proton hole transfer events follow until a new 'exposed' configuration is reached. This can occur in only two steps, that is, in a 'exposed'–'buried'–'exposed' or a 'tilted'–'buried'–'exposed' sequence (sketched in Fig. 4c,d, respectively). Yet, it can be the case that the 'exposed' configuration is not easily reachable, because a water molecule with a dangling bond exposed to the surface is not coordinated to the $OH^-$. Such situation, illustrated in Fig. 4e, leads to a superposition of 'tilted' and 'buried' structures. This effect imprints the shallow local free-energy minimum at $\delta \approx 0$ in the free-energy profile (Fig. 6a), which implies that the Zundel-like proton hole complex $[HO\ H\ OH]^-$ is not anymore a transient structure but rather an intermediate species. This trapping of the anionic Zundel complex due to nanoconfinement is in stark contrast to the transient Zundel-like transition state in homogeneous bulk environments as predicted by *ab initio* path integral simulations as a feature of the dynamical hypercoordination mechanism[40] and confirmed by time-resolved spectroscopy[41] for $OH^-(aq)$.

Finally, we provide some qualitative insights into both diffusion properties and reorientation behaviour (which are not directly accessible from our NVT simulations due to the required heavy thermostatting[33], see Methods). Towards structural diffusion of $OH^-(aq)$, we have computed conditional free-energy profiles where proton hole rattling has been approximately excluded; here, rattling events are simply identified by searching for proton hole transfer events of the kind $o_i^* \rightarrow o_j^* \rightarrow o_i^*$ and excluding the contribution from $O_j^*$ to the $P(\delta)$ probability distribution, to provide qualitative trends and, therefore, we do not claim these conditional profiles to be the proper free-energy profiles for proton (hole) diffusion. The resulting conditional free-energy barrier for system $N$, $\approx 1.3\ kcal\ mol^{-1}$, is a bit higher compared with that of the bulk and $W$ systems, being both $\approx 1.1\ kcal\ mol^{-1}$, according to Fig. 6b. Assuming that these conditional free-energy barriers are correlated with the diffusion coefficient of $OH^-(aq)$, they suggest that structural diffusion of the proton hole should be slightly less efficient in the narrow slit pore than in the $W$ system, which in turn is essentially identical to that in the respective bulk limit. Indeed, this inference for the $N$ system is in accord with the recent finding that $OH^-$ structural diffusion in a water monolayer confined between two graphene slabs is slowed down compared with the corresponding bulk environment as obtained from proper dynamical analyses[25].

Concerning the reorientation dynamics, it has been shown previously that there is a close connection between increased reorientation times and suppressed structural diffusion for hydroxide in bulk water, thus establishing a close link between proton (hole) transfer and $OH^-$ orientational relaxation[42]. To qualitatively disclose the relationship between these two processes in nanoconfined water in the absence of having access to any time evolution, we computed the probability distribution function for the reorientation angle on proton transfer, being similar in spirit to our indirect approach to the diffusion properties. Interestingly, the data in Fig. 7 unveil that the reorientation properties are remarkably different for system $N$ compared with system $W$ and bulk water. A proton hole transfer event has a probability of roughly 70% of causing a significant reorientation $>120°$ in system $N$, whereas that probability is only roughly 40% for the $W$ and bulk systems. This clearly reflects the peculiar migration mechanism sketched in Fig. 4: extreme confinement in system $N$ imposes that proton (hole) migration occurs via a zig-zag mechanism, which requires significant reorientational motion of

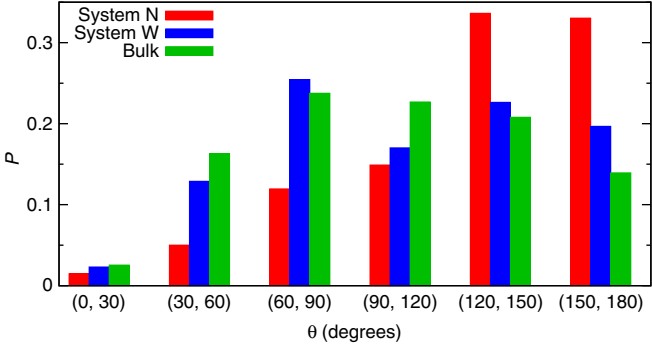

**Figure 7 | Probability distribution of the reorientation angle.** Probability distribution of the reorientation angle of the hydroxide anion on proton hole transfer defined in terms of the angle $\theta$ formed by the vectors along the $OH^-$ bond at time instants $t_i$ and $t_{i+1}$ immediately before and after a proton hole transfer event takes place.

the $OH^-$, whereas moderate confinement allows for smaller reorientations, in that sense being more similar to the bulk regime. Thus, the intimate connection between proton hole diffusion and $OH^-$ reorientation dynamics in the bulk[42] seems to hold also in nanoconfined water.

## Discussion

Our *ab initio* simulations of alkaline aqueous solutions subject to nanoconfinement reveal an unexpectedly rich dynamical landscape for proton hole transfer and thus for the migration of negative charge defects in terms of solvated hydroxide species, $OH^-(aq)$. Although the free-energy barriers for proton hole transfer in the narrow and wide slit pore systems N and W are remarkably similar, both to each other and compared with bulk water, the extracted mechanisms for structural diffusion of proton holes in these nanoconfined aqueous environments reveal major differences not only with reference to the bulk, but also between the two confinement scenarios. In particular, our results show that the diffusion regime is indeed qualitatively dependent on the degree of confinement, resulting in a monolayer or bilayer structuring of the water lamellae in the narrow and wide scenario, respectively. In the narrow pore, the interplay of nanoconfinement due to planar surfaces, fourfold hypercoordination and preferred perpendicular orientation of interfacial $OH^-(aq)$ species with respect to the surfaces, and the topology of the hydrogen-bonded water network that hosts the defect imprints a zig-zag charge migration pathway. As a result, the proton hole necessarily must jump between the two surfaces in each step, thereby reversing the orientation of the hydroxide's dipole moment, which should be detectable by suitable dielectric spectroscopy techniques.

Charge transport in the basic solution confined by a wider slit pore is not only distinctly different from the monolayer limit, but allows for a multitude of different migration and trapping mechanisms, which is traced back to the bilayer sub-structure of the confined water film in this case. First of all, the $OH^-(aq)$ defect can stay for a short time trapped at the interface in its preferred perpendicular hypercoordinated state due to a suppressed probability, to find a hydrogen bonded water molecule that can become a stable defect site after proton hole transfer, which results in pronounced rattling with respect to the trapped interfacial defect. Next, it is found that the hydroxide anion can be even stabilized in terms of an anionic Zundel-like intermediate with its characteristic centred hydrogen bond, $[HO\cdots H\cdots OH]^-$, instead of this Zundel complex being a

transition state and thus a transient species as encountered in bulk alkaline solutions. Last but not the least, several distinct charge transport scenarios have been observed that indeed lead to net defect displacements and thus contribute to long-range structural diffusion.

Capitalizing on all these observations, it should be possible to rationally design different nanostructures that will allow for different charge transport rates of confined alkaline aqueous solutions. This is much more promising compared with acidic solutions, as no differences regarding proton transfer or structural diffusion depending on the extent of nanoconfinement have been found for the hydrated excess proton $H^+(aq)$, observing that the standard Grotthuss diffusion mechanism is not at all hindered in any of the systems compared with the bulk. These facts put together open up very suggestive questions and ideas towards potential applications of nanoconfined alkaline water films in layered materials, especially concerning rational design of ion exchange membranes where the diffusion rate of $OH^-$ and thus charge transport can be tuned by controlling the degree of confinement imposed by the slit pores, while keeping the diffusion rate of $H^+$ unchanged.

In addition, our results are expected to add new insights, ideas and stimulation to the ongoing extensive debate about the molecular character of the water/air interface. Athough it is obviously out of the scope of this investigation—our system being subject to two-sided confinement inside a layered mineral—to directly address this controversy, we think that the now clearly revealed asymmetry between the diffusion mechanism of $OH^-$ versus $H^+$ species in interfacial water films can be a valuable piece of information for solving the puzzle of the origin of the observed charge in interfacial water.

Let us close our discussion by addressing more fundamental issues. Based on the multitude of predicted charge transport channels, including various forms of trapped defects, it is predicted that analyses of the long-time dynamics, which is accessible via empirical valence bond or neural network representations of $OH^-$ in water, will lead to the so-called 'broad' waiting time distributions in the sense of Lévy flights and non-Gaussian fluctuations. Here, systematic variation of the slit pore width will be the key control parameter that determines the particular dynamical scenario. Depending on that control parameter, it might be possible to switch in a well-defined manner between sub- and superdiffusive structural charge transport and thus to probe strongly fractional dynamics with long-memory effects.

## Methods

**Computational approach.** We employed the well-validated methods and system setups as in our previous studies of neutral and acidic water nanoconfined between mackinawite sheets[33,34]. We used the Perdew-Burke-Ernzerhof (PBE) density functional[43] with a plane wave cutoff of 25 Ry and ultrasoft pseudopotentials[44] containing *d*-projectors for sulfur and semicore states, as well as scalar relativistic corrections for iron. The *ab initio* molecular dynamics simulations[27] were carried out with the Car–Parrinello method as implemented in CPMD[45]. Massive Nosé–Hoover chain thermostats[46] were used for controlling the temperature of nuclei (at 500 K) and electrons, with a fictitious orbital mass of 700 a.u., a timestep of 2 a.u., substituting D for H masses and employing a very high-order Suzuki–Yoshida algorithm to properly integrate these thermostat equations of motion. Such unusually aggressive thermostatting was found to be necessary, to enforce stable Car–Parrinello propagation for these slit pore systems subject to metallic mineral confinement[33], which strictly prevents the calculation of time-correlation functions and thus of any dynamical properties. The systems were equilibrated for 5 ps, after which 40 ps of production runs were collected.

**Model systems.** We studied the alkaline solution in 'extreme' and 'moderate' nanoconfinements due to mackinawite sheets[33,34] denoted as the narrow and wide slit pore systems N and W, respectively. They consist of two $Fe_{32}S_{32}$ parallel layers situated at the top and bottom of a tetragonal supercell with $a = b = 14.69$ Å. The ideal spacing of 5.03 Å is kept between the top and bottom layers of the two distinct

mineral sheets, where the top- and bottom-most S atoms of the top and bottom layers are frozen at their ideal crystal positions, thus keeping all atoms at the water/mineral interface mobile. In system $N$, $c = 13.82$ Å and the interlayer space is filled with 31 $H_2O$ molecules plus 1 $OH^-$, whereas in system $W$, $c = 16.73$ Å and the interlayer contains 48 $H_2O$ molecules plus 1 $OH^-$. This roughly corresponds to an estimated pressure of $\approx 20$ MPa at $T = 500$ K, thus approximately mimicking elevated temperature and pressure conditions. For the corresponding reference simulations for $OH^-$ in homogeneous bulk water, we used accordingly a cubic cell of $a = 10.415$ Å containing 31 $H_2O$ and 1 $OH^-$ unit.

**Validation of the approach.** The computational approach and model setup have been extensively validated in our previous studies[33,34]. In particular, it was checked[34] that the fictitious orbital mass yields stable Car–Parrinello propagation and tests with both lower and higher values (of 500 and 900 a.u.) yielded similarly stable integration. We note that even though it is possible to find some debate in the literature concerning possible artefacts in non-dynamical properties as a consequence of using large fictitious orbital masses in Car–Parrinello simulations, these concerns have already been properly addressed (as discussed for instance in Section 2.4.9 of ref. 27). Another note concerns our choice of the exchange-correlation functional. Plain PBE has been shown to broadly yield accurate and robust results without the need of adding dispersion corrections and in particular for hydrogen-bonded systems the inclusion of (D2 or D3) dispersion corrections has been demonstrated to even significantly increase the average error[47]. In addition, we carefully checked that the plain PBE functional generates the correct dynamical hypercoordination mechanism[1] for structural diffusion of $OH^-$ in bulk water at ambient conditions. Moreover, on increasing temperature and pressure of bulk water to 500 K and $\approx 20$ MPa, the essential features of this mechanism were observed to remain unchanged based on carefully analysing the active and resting states in terms of conditional radial distribution functions and the corresponding running coordination numbers in the limits $\delta < 0.1$ and $\delta < 0.5$ Å, respectively. The only difference worth noting was a slight weakening of the weak hydrogen bond that can be donated via H′ in the active state as revealed by a shoulder instead of a small peak at $\approx 2$ Å. Finally, bonding charge analyses carried out for alkaline solutions revealed no net charge transfer between the mineral layers and the liquid phase across the interface, which is consistent with previous findings for the nanoconfined excess proton[34].

**Data availability.** The data that support the findings of this study are available from the corresponding author upon request.

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

## Acknowledgements

This work has been supported by the German Research Foundation (DFG) via the Cluster of Excellence EXC 1069 'RESOLV'. We also gratefully acknowledge the Gauss Centre for Supercomputing (GCS) for providing computing time for a GCS Large Scale Project on the IBM Blue Gene/Q system Juqueen[48] at Jülich Supercomputing Centre (JSC) as well as HPC-RESOLV and BOVILAB@RUB.

## Author contributions

D.M.-S. performed the simulations. D.M.-S. and D.M. analysed the results and wrote the manuscript.

## Additional information

**Competing financial interests:** The authors declare no competing financial interests.

