## [Peer Review File · Nature Communications]

Reviewers' comments:

Reviewer #1 (Remarks to the Author):

This is a very interesting and timely communication on the dynamics of the OH⁻ ion transport in nano-confined water between corrugated mackinawite FeS sheets that the authors have studied using ab-initio molecular dynamics (CP-MD) simulations. The results quite are different from their previous studies of hydrated excess proton transport (reference 32) in the same systems, where no major differences were found in the mechanism of proton transport in mono or bilayer water sandwiched between the sheets. In contrast to this, the hyper-coordinated solvation OH⁻ (H₂O)₄ of the hydroxide ion, the topology of the hydrogen bond network enable different mechanisms of transport in mono and bilayer water sheets. The OH⁻ ion prefers to be oriented perpendicular to the mackinawite-water interface in the monolayer; switching from one face to the other during the structural diffusion of the proton hole while several other options are available for transport in the water bilayer in which the OH⁻ can assume in addition buried and tilted conformations and others in which a Zundel intermediate may also be stabilized.

The work is original and the quantum simulations have been carefully carried out by experts in the field.

The findings should be important in the design layered nano-membranes that can be tuned to permit different rates of transport of OH ions, of interest in the design of alkaline fuel cells and alkaline exchange membranes as alternatives to proton fuel cells and proton exchange membranes. The asymmetry in the diffusion of H⁺ and OH⁻ ions should help elucidate the puzzle of whether the air-water interface is basic or acidic and in understanding the possible role of Mackinawite as a nano-reactor and catalyst in prebiotic chemistry (reference 28).

I ask for one clarification- does the OH⁻ ion move faster or slower in the bilayer than in the monolayer or are they about the same? (see fig 6 for the free energy profile for proton hole transfer) . Presumably it is slower than in the one dimensional water wires but this should be stated.

The unusual behavior of the hydroxide ion in nanoconfined water is of wide interest and publication is strongly recommended.

Reviewer #2 (Remarks to the Author):

In this article, the authors employ ab initio molecular dynamics (AIMD) to study the solvation and transport of the hydroxide (OH⁻) ion in a nanoconfined aqueous environment. Previously, the AIMD protocol used in this study had been successfully employed predict the transport mechanism of OH⁻ in bulk water, and the same protocol is employed in this study. In this work, the authors confine the hydroxide ion in mackinawite slits, one of which is a narrow slit and the other, a wide slit so as to produce two different nanoconfinement conditions. Details of the change in the solvation structure and transport mechanism compared to the bulk are revealed in the calculations. The solvation structures obtained are of three types: Exposed, buried, and tilted, all of which influence the transport mechanism in a particular way.

This work is quite impressive and will likely be of broad interest to the physical chemistry community. I would like to point out a few issues that could use some clarification in a revised version of the paper.

1. The connection of this work to experiment is rather weak, and therefore, it is unclear why the authors chose to perform their simulations at 500 K, and no explanation is given. What is the

physical basis for this choice? Moreover, given that the high temperature is also likely to influence the OH⁻ transport mechanism in these systems, is a comparison to bulk at 300 K really valid?

2. It is something of a shame that the authors chose to perform heavily thermostatted molecular dynamics, as this scheme necessarily destroys dynamical properties, several of which would have been particularly useful to examine in this study. As examples, insights from hydroxide diffusion constants and reorientation times would have added to the richness of the results of this study. Although these properties are unavailable from the AIMD trajectories generated by the authors, I think some comment on what they might expect for these quantities would be helpful. As a point of comparison, I would refer them to an AIMD study of OH⁻ reorientation times, and their connection to structural diffusion, in bulk by Ma and Tuckerman [Chem. Phys. Lett. 511, 177 (2011)].

3. I would like to see more discussion about the hydrogen bond donated by OH⁻ via H⁺, particularly in the W channel, as this donor bond plays an important role in the transport mechanism when it can form. It seems that it would be most likely to form in the B and T configurations. Is this the case? Does it influence structural diffusion in the same way as it does in bulk transport? These are questions that seem to be especially relevant and should be given a bit more space in the paper.

Point-by-Point Reply to the Reviewers' Questions and Suggestions

Note: Changes in the revised Manuscript are **highlighted in red**; in addition we changed the title of the Manuscript in order to comply with *Nature Communications*' guidelines.

REVIEWER #1

This is a very interesting and timely communication on the dynamics of the OH⁻ ion transport in nano-confined water between corrugated mackinawite FeS sheets that the authors have studied using ab-initio molecular dynamics (CP-MD) simulations. The results quite are different from their previous studies of hydrated excess proton transport (reference 32) in the same systems, where no major differences were found in the mechanism of proton transport in mono or bilayer water sandwiched between the sheets. In contrast to this, the hyper-coordinated solvation OH⁻ (H₂O)₄ of the hydroxide ion, the topology of the hydrogen bond network enable different mechanisms of transport in mono and bilayer water sheets. The OH⁻ ion prefers to be oriented perpendicular to the mackinawite-water interface in the monolayer; switching from one face to the other during the structural diffusion of the proton hole while several other options are available for transport in the water bilayer in which the OH⁻ can assume in addition buried and tilted conformations and others in which a Zundel intermediate may also be stabilized.

The work is original and the quantum simulations have been carefully carried out by experts in the field.

The findings should be important in the design layered nano-membranes that can be tuned to permit different rates of transport of OH ions, of interest in the design of alkaline fuel cells and alkaline exchange membranes as alternatives to proton fuel cells and proton exchange membranes. The asymmetry in the diffusion of H⁺ and OH⁻ ions should help elucidate the puzzle of whether the air-water interface is basic or acidic and in understanding the possible role of Mackinawite as a nano-reactor and catalyst in prebiotic chemistry (reference 28).

The unusual behavior of the hydroxide ion in nanoconfined water is of wide interest and publication is strongly recommended.

Point 1

I ask for one clarification- does the OH⁻ ion move faster or slower in the bilayer than in the monolayer or are they about the same? (see fig 6 for the free energy profile for proton hole transfer). Presumably it is slower than in the one dimensional water wires but this should be stated.

Reply: Our mineral confinement is, unfortunately, in a metallic electronic state. This has been carefully analyzed in Ref. 1, see in particular Section III.A therein. As a consequence, we have to enforce Car-Parrinello adiabaticity [2] of the mineral/water slit pore by most aggressively thermostating both the nuclei and orbitals, as discussed in Section II.B, in terms of massive thermostating using unusually long Nosé-Hoover chains and very tight Suzuki-Yoshida integration. This well-tested procedure resulted in stable Car-Parrinello propagation that produced correct equilibrium fluctuations.

Due to the strict requirement of heavily thermostating the composite system – switching off the thermostats resulting in both immediate and severe CP nonadiabaticity and energy flow – we have no access to quantities such as diffusion coefficients (see also our reply to Point 2 of Reviewer #2). Nevertheless, we can provide interesting insights into the posed question (if the OH⁻ ion diffuses faster or slower in the bilayer than in the monolayer) by inspecting the free energy profiles for proton hole transfer which result from the underlying proton hole dynamics. In the original Manuscript, the ΔF profiles in Fig. 6 are computed employing the whole AIMD trajectory, which implies that these profiles include not only proton transfer events that lead to structural diffusion of the OH⁻ unit and thus proton transport, but also 'proton rattling' events involving locally pinned defects that do not contribute to the structural diffusion of OH⁻. Stimulated by the Reviewer's request we have reanalyzed these profiles by excluding such rattling events using a very simple tagging approach that does not attempt at being quantitative in any respect (see Fig. 1); note that we therefore do not claim these conditional profiles to be the proper free energy profiles for proton (hole) diffusion. As it is clearly seen, this conditional free energy barrier is slightly higher in the N system than in the W system, which in turn is essentially identical to that in the bulk. This implies not only that the OH⁻ diffusion should be faster in the W system, but also that similar diffusion coefficients could be expected for the bulk and W systems. Indeed, this inference for the N system is in accord with the recent observation of Bankura and

FIG. 1. Free energy profiles for proton transfer in the N and W systems compared to the bulk environment, both including proton rattling events (left panel, reproduced from the original Manuscript) or excluding them (right panel, new data). Rattling events are excluded by searching for proton transfer events of the kind $O_i^* \rightarrow O_j^* \rightarrow O_i^*$ and ignoring the contribution from O_j^* to the $P(\delta)$ probability distribution.

Chandra that OH^- diffusion in a water monolayer confined between two graphene slabs is slowed down compared to the corresponding bulk environment [3].

We now refer to the thermostating issue in the Methods section and discuss the new insights into the diffusion coefficient in a separate paragraph that has been added toward the end of the Results section of the revised Manuscript, which includes an amended Figure 6 (b). We note that similar analysis of reorientational motion has been added to the Results section in response to Reviewer #2.

REVIEWER #2

In this article, the authors employ ab initio molecular dynamics (AIMD) to study the solvation and transport of the hydroxide (OH^-) ion in a nanoconfined aqueous environment. Previously, the AIMD protocol used in this study had been successfully employed predict the transport mechanism of OH^- in bulk water, and the same protocol is employed in this study. In this work, the authors confine the hydroxide ion in mackinawite slits, one of which is a narrow slit and the other, a wide slit so as to produce two different nanoconfinement conditions. Details of the change in the solvation structure and transport mechanism compared to the bulk are revealed in the calculations. The solvation structures obtained are of three types: Exposed, buried, and tilted, all of which influence the transport mechanism in a particular way.

This work is quite impressive and will likely be of broad interest to the physical chemistry community. I would like to point out a few issues that could use some clarification in a revised version of the paper.

Point 1

The connection of this work to experiment is rather weak, and therefore, it is unclear why the authors chose to perform their simulations at 500 K, and no explanation is given. What is the physical basis for this choice? Moreover, given that the high temperature is also likely to influence the OH^- transport mechanism in these systems, is a comparison to bulk at 300 K really valid?

Reply: The present work is part of a broad, long-term investigation on prebiotic peptide synthesis [4–6] in nanoconfined water between mackinawite FeS sheets [1, 7] (where currently only some preliminary data on reactions are available [8]). Why this particular mineral? It has been established that mackinawite can be produced at hydrothermal vent conditions [9], and recently it has been pointed out to be one of the possible key catalysts for the emergence of

life in such conditions [10]. Particularly, water-filled slit nanochannels formed upon intercalation of water between mackinawite sheets have also been proposed as part of primordial ‘pyrophosphate-synthetase nanoengines’ [11, 12]. In this context, we set out to explore the effect of nanoconfinement on several chemical reactions leading to prebiotic peptide synthesis *at hydrothermal vent conditions*, i.e. in nanoconfined water at $T = 500$ K and $p \sim 20$ MPa. As a necessary prologue to that investigation, we studied the properties of both neutral water [1] and acidic water [7] at high temperature and pressure confined in mackinawite sheets, and in the same spirit we carried out the present study. It is for these reasons that we performed our simulations at such extreme thermodynamic conditions.

In response to this question, the third paragraph in the Introduction has been amended, including additional references to relevant prebiotic work, in order to clearly expose this physical basis for our choice.

FIG. 2. Radial distribution functions $g(r)$ and running coordination numbers $N(r)$ for the pairs $\text{H}'\text{O}$ and O^*H . Note that the former includes all oxygens in the system and the latter includes all hydrogens.

Regarding the requested comparison with ambient bulk, note that we refer in the Manuscript to a simulation of OH^- (aq) in bulk water at extreme conditions ($T = 500$ K, $p \sim 20$ MPa), but in order to validate our approach we also performed a simulation of OH^- (aq) in ambient bulk water. As already exposed in the original main text, we observed the proper structural diffusion of the OH^- (aq) in the bulk taking place by the dynamical hypercoordination mechanism [13]. In particular, our results for ambient bulk water closely resemble those described in the literature obtained with the BLYP functional [13], while upon increasing temperature and pressure the changes are purely quantitative without affecting the basic features of the migration mechanism itself. This can be easily checked by

examination of the radial distribution function for the pairs H'O and O*H as shown in Fig. 2 for the two bulk simulations, i.e. at ambient conditions and at high temperature and pressure being relevant to the present study.

While the 'resting' configurations for the OH⁻(aq) far from proton transfer events are clearly hypercoordinated at both ambient and extreme conditions (as shown by the $N_{\text{O}^*\text{H}} \approx 5$ value at $r \approx 2.5$ Å for $\delta > 0.5$), the 'active' configurations ($\delta < 0.1$) are tetrahedrally coordinated as shown by the smaller $N_{\text{O}^*\text{H}}$ value of ≈ 4 at the same r ; note that the hydroxide's H site is included in defining the coordination number of O*. The only appreciable detail in this case when comparing ambient and extreme conditions is the very small displacement of the hydrogen bond donating water in the first solvation shell of the OH⁻ towards higher values of r and a small loss of the ideal square-planar structure as expected when rising temperature to 500 K. On the other hand, the hydrogen bond donated by the OH⁻ via its free hydrogen H' is non-existent at both ambient and extreme conditions when $\delta > 0.5$, while a weak donor bond is formed close to proton transfer events. This is quite pronounced for ambient bulk water (black line), as shown by the corresponding small maximum of $g_{\text{H}'\text{O}}$ at $r = 2$ Å, while at extreme conditions (green line) this peak is washed out to a shoulder at the same r , as expected at higher temperatures according to its already weak character in ambient water. While this points towards a less important role of the donated H' bond in the proton hole transfer mechanism at high temperatures, the clear differences in both $g(r)$ and $N(r)$ between the active and resting configurations can be still appreciated at extreme conditions akin to ambient. Hence, we conclude that the essential features of the proton (hole) transfer mechanism are conserved when changing from ambient to extreme conditions.

We now explain these important observations in the revised Manuscript when discussing the role of the H' bond in the dynamical hypercoordination mechanism in Results C and in the Methods section when we refer to the bulk reference simulations in the Validation part.

Point 2

It is something of a shame that the authors chose to perform heavily thermostatted molecular dynamics, as this scheme necessarily destroys dynamical properties, several of which would have been particularly useful to examine in this study. As examples, insights from hydroxide diffusion constants and reorientation times would have added to the richness of the results of this study. Although these properties are unavailable from the AIMD trajectories generated by the authors, I think some comment on what they might expect for these quantities would be helpful. As a point of comparison, I would refer them to an AIMD study of OH⁻ reorientation times, and their connection to structural diffusion, in bulk by Ma and Tuckerman [Chem. Phys. Lett. 511, 177 (2011)].

Reply: Indeed, the calculation of reorientation times or diffusion constants would have been a wonderful complement to our results! Unfortunately, the metallic character of the mackinawite slabs makes it impossible to carry out any simulation in the NVE ensemble since we have to apply unusually heavy thermostating to enable stable Car-Parrinello propagation. This has been carefully analyzed in Ref. 1, see in particular Section III.A therein. As a consequence, we have to enforce Car-Parrinello adiabaticity [2] of the mineral/water slit pore by most aggressively thermostating both the nuclei and orbitals, as discussed in Section II.B in Ref. 1, in terms of massive thermostating using unusually long Nosé-Hoover chains and very tight Suzuki-Yoshida integration. This prevents any extraction of time information and thus of dynamical properties.

This technical background is now clarified in the revised Methods section.

However, due to the strong stimulation of Reviewer #2, we have tried our best to extract some insights regarding a possible link [14] between structural diffusion and of OH⁻ reorientation based on equilibrium information. Concerning diffusion, we kindly refer to our feedback to Reviewer #1 where we analyzed conditional free energy profiles for proton hole transfer by excluding proton rattling events using a simplistic criterion in order to provide qualitative trends. Concerning OH⁻ reorientation, while we have not access to real-time information and thus cannot compute the proper $C_2(t)$ time-correlation functions, we can compute the probability distribution for the reorientation angle upon proton transfer, which is the angle θ formed between the OH⁻ units at instants t_i and t_{i+1} , corresponding respectively to the situations immediately before and after a proton transfer takes place (Fig. 3).

It is revealed that the reorientation in system N is distinctly different from that in the wide pore and in the bulk, while these two are not qualitatively different. As explained in more detail in the revised manuscript, this is consistent with the diffusion behavior that is suggested by the conditional free energy barriers, which are again similar for system W and bulk but different from that in the narrow pore. Thus, also in nanoconfinement there appears to be a correlation between hole migration and orientational relaxation akin to what has been shown previously for the bulk solution by Ma and Tuckerman [14].

Our additional reorientational analysis and the finding that structural diffusion and reorientational

FIG. 3. Probability distribution of the reorientation angle upon proton transfer θ , i.e. the angle formed between the OH^- vectors immediately before and after a proton transfer event takes place.

motion are connected has been introduced in the revised Manuscript as the new last paragraph of the Results section, including the associated new Figure 7. We note that the analogous analysis of the structural diffusion properties has been added to the Results section in the previous paragraph already in response to Reviewer #1.

Point 3

I would like to see more discussion about the hydrogen bond donated by OH^- via H' , particularly in the W channel, as this donor bond plays an important role in the transport mechanism when it can form. It seems that it would be most likely to form in the B and T configurations. Is this the case? Does it influence structural diffusion in the same way as it does in bulk transport? These are questions that seem to be especially relevant and should be given a bit more space in the paper.

Reply: As we already clarified in our response to the previous Point 1, this H' donor bond found for ‘active’ configurations is a bit less defined in hot–pressurized bulk water compared to ambient conditions. In particular, the small $g_{\text{H}'\text{O}}$ peak at $\approx 2 \text{ \AA}$ transmutes into a shoulder at high temperature and pressure bulk conditions, see Fig. 2, thus it is still present but to a lesser extent. We may infer for the nanoconfined solution that the formation of this specific bond, being apparently not so important at extreme bulk conditions, plays now a less crucial role in the nanoconfined proton hole transfer process. Indeed, refined analysis of the $g(r)$ and $N(r)$ functions split into the $\delta < 0.1$ and $\delta > 0.5$ complexes for the N system (Fig. 2) and for the E configurations in the W system (Fig. 4) reveals that this H' bond is completely absent in these two cases, while it is even more pronounced compared to the high temperature/pressure bulk reference for T/B configurations in the W system in the ‘active’ state ($\delta < 0.1$); this donated hydrogen bond is always absent in the ‘resting’ state ($\delta > 0.5$) consistent with the expectation. This fully confirms the suggestion of the Reviewer.

Looking now at the full picture based on this comprehensive additional analysis, it seems clear that while the formation of this donor bond is not a necessary condition for proton hole transfer to occur at the studied conditions, its more pronounced presence nicely correlates with the free energy barrier for proton hole transfer being small (as beautifully seen for configurations T/B compared to E in system W). Hence, at the studied conditions, including nanoconfinement, the formation of this donor bond is not strictly required for proton hole transfer but when it exists it greatly favors the transfer process, or using the Reviewer’s words: “*this donor bond plays an important role in the transport mechanism when it can form*”.

These conclusions have been added to the Manuscript in the revised Results C section when discussing the transfer properties related to T/B versus E configurations in system W.

FIG. 4. Radial distribution functions $g(r)$ and running coordination numbers $N(r)$ for the pairs $\text{H}'\text{O}$ in the W system resolved into ‘Exposed’, ‘Tilted’ and ‘Buried’ configurations for active (left panel) and resting (right panel) states together with the corresponding data for the bulk system for comparison.

-
- [1] Wittekindt, C. & Marx, D. Water confined between sheets of mackinawite FeS minerals. *J. Chem. Phys.* **137**, 054710 (2012).
 - [2] Marx, D. & Hutter, J. *Ab Initio Molecular Dynamics: Basic Theory and Advanced Methods* (Cambridge University Press, Cambridge, 2009).
 - [3] Bankura, A. & Chandra, A. Proton transfer through hydrogen bonds in two-dimensional water layers: A theoretical study based on ab initio and quantum-classical simulations. *J. Chem. Phys.* **142**, 044701 (2015).
 - [4] Pollet, R., Boehme, C. & Marx, D. Ab initio simulations of desorption and reactivity of glycine at a water-pyrite interface at “Iron-Sulfur world” prebiotic conditions. *Origins Life Evol. B.* **36**, 363–379 (2006).
 - [5] Schreiner, E., Nair, N. N. & Marx, D. Influence of extreme thermodynamic conditions and pyrite surfaces on peptide synthesis in aqueous media. *J. Am. Chem. Soc.* **130**, 2768–2770 (2008).
 - [6] Schreiner, E., Nair, N. N., Wittekindt, C. & Marx, D. Peptide synthesis in aqueous environments: the role of extreme conditions and pyrite mineral surfaces on formation and hydrolysis of peptides. *J. Am. Chem. Soc.* **133**, 8216–8226 (2011).
 - [7] Muñoz-Santiburcio, D., Wittekindt, C. & Marx, D. Nanoconfinement effects on hydrated excess protons in layered materials. *Nat. Commun.* **4**, 2349 (2013).
 - [8] Muñoz-Santiburcio, D. & Marx, D. Prebiotic chemistry in nanoconfinement. In *NIC Symposium 2016 – Proceedings*, 125–132 (NIC, FZ Jülich, 2016).
 - [9] Mielke, R. E. *et al.* Iron-Sulfide-Bearing Chimneys as Potential Catalytic Energy Traps at Life’s Emergence. *Astrobiology* **11**, 933–950 (2011).
 - [10] White, L. M., Bhartia, R., Stucky, G. D., Kanik, I. & Russell, M. J. Mackinawite and greigite in ancient alkaline hydrothermal chimneys: identifying potential key catalysts for emergent life. *Earth Planet. Sci. Lett.* **430**, 105–114 (2015).
 - [11] Russell, M. J., Nitschke, W. & Branscomb, E. The inevitable journey to being. *Phil. Trans. R. Soc. B* **368**, 20120254 (2013).
 - [12] Russell, M. J. *et al.* The drive to life on wet and icy worlds. *Astrobiology* **14**, 308–343 (2014).
 - [13] Marx, D., Chandra, A. & Tuckerman, M. E. Aqueous basic solutions: hydroxide solvation, structural diffusion, and comparison to the hydrated proton. *Chem. Rev.* **110**, 2174–2216 (2010).
 - [14] Ma, Z. & Tuckerman, M. E. On the connection between proton transport, structural diffusion, and reorientation of the hydrated hydroxide ion as a function of temperature. *Chem. Phys. Lett.* **511**, 177–182 (2011).

REVIEWERS' COMMENTS:

Reviewer #1 (Remarks to the Author):

I am satisfied with the response of the authors to my request for clarification of the relative rates of diffusion of the OH⁻ in the water bilayer and water monolayer between corrugated mackinawite sheets. I have read the revised paper and accept all the other changes. The paper is very interesting and should be published.

Minor corrections:

Page 6 second para line 4 - change "termostatting" to "thermostatting"

Reference 24: change h⁺ and oh⁻ ions to "H⁺ and OH⁻ ions"

Reviewer #2 (Remarks to the Author):

I have read the revised manuscript and responses of the authors, and I believe they have adequately addressed the concerns raised in my report. The manuscript is, in my view, now suitable for publication in Nature Communications.

Point-by-Point Reply to the Reviewers' Questions and Suggestions (2nd Revision)

REVIEWER #1

I am satisfied with the response of the authors to my request for clarification of the relative rates of diffusion of the OH⁻ in the water bilayer and water monolayer between corrugated mackinawite sheets. I have read the revised paper and accept all the other changes. The paper is very interesting and should be published.

Minor corrections:

Page 6 second para line 4 - change "termostatting" to "thermostatting"

Reference 24: change h⁺ and oh⁻ ions to "H⁺ and OH⁻ ions"

REVIEWER #2

I have read the revised manuscript and responses of the authors, and I believe they have adequately addressed the concerns raised in my report. The manuscript is, in my view, now suitable for publication in Nature Communications.

Reply: We are most thankful for the Reviewers' comments and feedback. We have addressed the minor corrections suggested by Reviewer #1.